# Extracorporeal Shock Wave Therapy Salvages Critical Limb Ischemia in B6 Mice through Upregulating Cell Proliferation Signaling and Angiogenesis

**DOI:** 10.3390/biomedicines10010117

**Published:** 2022-01-06

**Authors:** Pei-Hsun Sung, Tsung-Cheng Yin, Han-Tan Chai, John Y. Chiang, Chih-Hung Chen, Chi-Ruei Huang, Hon-Kan Yip

**Affiliations:** 1Division of Cardiology, Department of Internal Medicine, Kaohsiung Chang Gung Memorial Hospital, Chang Gung University College of Medicine, Kaohsiung 833253, Taiwan; e12281@cgmh.org.tw (P.-H.S.); chaiht@mail.cgmh.org.tw (H.-T.C.); starchmay33@gmail.com (C.-R.H.); 2Center for Shockwave Medicine and Tissue Engineering, Kaohsiung Chang Gung Memorial Hospital, Kaohsiung 833253, Taiwan; 3Institute for Translational Research in Biomedicine, Kaohsiung Chang Gung Memorial Hospital, Kaohsiung 833253, Taiwan; 4Department of Orthopedics, Kaohsiung Chang Gung Memorial Hospital, Chang Gung University College of Medicine, Kaohsiung 833253, Taiwan; tsongchenyin@gmail.com; 5Department of Computer Science and Engineering, National Sun Yat-Sen University, Kaohsiung 804201, Taiwan; chiang@cse.nsysu.edu.tw; 6Department of Healthcare Administration and Medical Informatics, Kaohsiung Medical University, Kaohsiung 807378, Taiwan; 7Divisions of General Medicine, Kaohsiung Chang Gung Memorial Hospital, Chang Gung University College of Medicine, Kaohsiung 833253, Taiwan; totoro@cgmh.org.tw; 8Department of Medical Research, China Medical University Hospital, China Medical University, Taichung 404333, Taiwan

**Keywords:** critical limb ischemia, extracorporeal shock wave, cell proliferation, cell growth and cell motility signalings

## Abstract

(1) This study tests hypothesis whether extracorporeal shock wave (ECSW) therapy effectively salvages mouse critical limb ischemia (CLI). In vitro result demonstrated that the angiogenesis parameters (i.e., tubular length/cluster/network formation) and protein expressions of EGFR/VEGFR2/RAS/c-Raf/MEK/ERK/VEGF/p-PI3K/p-Akt/p-m-TOR were significantly and progressively increased with stepwise augmentation of ECSW energy (0.1/0.14/0.20 mJ/mm^2^/140 impulses). On the other hand, they were suppressed by administration of Avastin (20 μM). Adult male B6 mice (*n* = 24) were equally categorized into group 1 (sham-operated control), group 2 (CLI), group 3 [CLI + ECSW (0.12 mJ/mm^2^/120 impulses/at days 1/3/7 after CLI induction)] and group 4 [CLI + ECSW (0.12 mJ/mm^2^/120 impulses) + Avastin (1 mg/intramuscular-injection)] at days 1/3/7 after CLI induction] and quadriceps were harvested by day 14. The laser Doppler result showed that the ratio of left (ischemia) to right (normal) limb blood flow was highest in group 1, lowest in group 2, and significantly higher in group 3 than in group 4 by days 7/14 after the CLI procedure (*p* < 0.0001). The protein expressions of cell proliferation/migration/angiogenesis receptors (EGFR/VEGFR2), angiogenesis biomarkers (VEGF/CXCR4/SDF-1) and cell proliferation/growth/survival (Ras/c-Raf/MEK/ERK)/(PI3K/Akt/m-TOR) and cell motility/proliferation (p-FAK/p-Scr) signaling biomarkers were significantly higher in group 3 than in groups 1/2/4, and significantly lower in group 1 than in groups 2/4, but they did not show a difference between groups 2 and 4 (all *p* < 0.001). The small vessel density and cellular levels of endothelial cell surface marker (CD31+) exhibited an identical pattern of blood flow, whereas the angiogenesis (CXCR4+/VEGF+) displayed an identical pattern of VEGFR2 among the groups (all *p* < 0.0001). The in vitro and in vivo studies found ECSW salvaged the CLI mainly through upregulating Ras-Raf-MEK/ERK/cell motility, cell proliferation/growth pathways and angiogenesis.

## 1. Introduction

Peripheral arterial disease (PAD) is not only a major predictor of atherosclerotic cardiovascular disease [1,2] but is also accompanied with multiple comorbidities [3], leading to a high risk of amputation and mortality [4]. Epidemiological and clinical observational studies [5,6] have found that about 10% of the population less than 70 years in age, 15–20% of those aged 70–85 years and 50% in those aged more than 85 years have PAD, highlighting that PAD is much more common in the elderly. In our clinical practice, the estimated prevalence of either asymptomatic or symptomatic PAD was up to 13% in people aged over 50 years old [5]. Additionally, about one in three symptomatic patients will eventually progress into intermittent claudication or critical limb ischemia (CLI) [7]. Furthermore, more than 70% of PAD results from poorly controlled diabetes and chronic kidney disease [8,9]. Therefore, the socioeconomic and medical burdens from either PAD per se or its associated comorbidities are considerable [10].

PAD/CLI treatment remains a great challenge to surgeons and clinicians [11]. In addition to standard anti-platelet and anti-ischemic medical treatments [12,13], percutaneous transluminal angioplasty and peripheral bypass surgery are currently two common therapeutic procedural strategies for PAD/CLI [14,15]. However, owing to the high incidence of atherosclerotic restenosis, the victims of PAD/CLI often require repeated percutaneous transluminal angioplasty, secondary bypass surgery, or eventually amputation for severe CLI [16]. These could explain why the long-term outcomes remain regrettably unfavorable in these PAD/CLI patients [17,18]. In view of the lack of an effective treatment for PAD/CLI patients, the development of an innovative treatment for these disease entities is urgent and of paramount importance.

Extracorporeal shock wave (ECSW) treatment has been well documented as an effective and safe noninvasive therapy for musculoskeletal or non-musculoskeletal disorders [19,20,21], mainly through suppressing the inflammatory process, reducing painful sensations and strengthening tissue repair. Our preclinical studies have previously demonstrated that ECSW therapy effectively ameliorated ischemic related organ dysfunction [22,23,24,25,26,27] mainly through enhancing vasculogenesis and neovascularization, mobilizing EPCs from bone marrow into circulation and homing to ischemic organ/tissues for angiogenesis, upregulating the expression of SDF-1α in ischemic zones for tracking the CXCR4+ cells into the ischemic area for angiogenesis and suppressing inflammatory reaction in ischemic and peri-infarcted zones. However, the deep and exact underlying mechanism for how the ECSW therapy would salvage the CLI is still currently unclear.

A body of basic research [28,29] has identified that epithelial growth factor receptor (EGFR) plays a crucial role for initiation and upregulation of Ras-Raf-MEK-ERK signaling for cell proliferation and survival. Additionally, the cell proliferation, growth and survival pathways of PI3K/Akt/m-TOR are frequently activated by ischemic stimulation [30], and the p-FAK/p-Scr pathway plays an essential role in cell motility and proliferation [31]. On the other hand, our previous study showed that ECSW therapy upregulated angiogenesis through activation of VEGFR2 [32]. Therefore, we proposed that the way that ECSW therapy salvaged the CLI in rat might be through regulating the angiogenesis, cell proliferation/growth/survival and cell motility signalings.

## 2. Materials and Methods

### 2.1. Ethics Statement

Experimental protocols were approved by the Institutional Animal Care and Use Committee at Kaohsiung Chang Gung Memorial Hospital (Affidavit of Approval of Animal Use Protocol No. 2019032008). Animals were housed in an Association for Assessment and Accreditation of Laboratory Animal Care International-approved animal installation in our institute, with standard temperature and light cycles.

### 2.2. In Vitro Study for Assessment of ECSW Therapy on Upregulating the Protein Expression of (1) Cell-Proliferation/Growth/Survival and (2) Cell-Motility Signaling Pathways

To facilitate understanding of the mechanism of how ECSW therapy would enhance the angiogenesis, cell survival and salvage of CLI, an in vitro study using human umbilical vein endothelial cells (HUVECs) was utilized. The HUVECs were categorized into group A [HUVECs (i.e., as control group)], group B [HUVECs + ECSW (energy stepwise increased from 0.1, 0.14 to 0.2 mJ/mm^2^/140 impulses)], and group C [HUVECs + ECSW/0.20 mJ/mm^2^/140 impulses) + Avastin 20 μM (Bevacizumab, an angiogenic monoclonal antibody)]. The dosage of Avastin for the in vitro study was based on the previous report [33] with some modification.

### 2.3. Animal Model of CLI, Animal Grouping, and Treatment Strategy

The methodologies have been reported in our previous studies [34,35]. In detail, C57BL/6J (B6) male mice (Charles River Technology, BioLASCO, Taiwan) were anesthetized by inhalation of 2.0% isoflurane. Under sterile conditions, the left femoral artery, small arteries, and circumferential femoral artery were exposed and ligated over their proximal and distal portions prior to removal. For mice that served as sham-operated control group, the arteries were only separated without ligation.

For the purpose of the study, animals were categorized into four groups: (sham-operated control), group 2 (CLI), group 3 [CLI + ECSW (0.12 mJ/mm^2^/120 impulses/at days 1, 3 and 7 after CLI induction)] and group 4 [(CLI + ECSW (0.12 mJ/mm^2^/120 impulses) and Avastin (1 mg/intramuscular-injection) at days 1, 3 and 7 after CLI induction]. The ECSW energy applied to mice was based on our previous reports with some modifications [23,30]. The dosage of Avastin for the in vivo study was based on a previous report [36] with some modification.

### 2.4. Measurement of Blood Flow in CLI by Laser Doppler

The methodologies have been reported in our previous studies [34,35]. In detail, mice were anesthetized by inhalation of isoflurane (2.0%) before CLI induction and at days 1, 7, 14 after CLI induction. The mice were placed on supine position on a warming pad (37 °C) and blood flow was measured in both inguinal areas using a laser Doppler scanner (moorLDLS, Moor Instruments, UK). The ratio of blood stream in the left (ischemic) leg and right (normal) leg was computed. On day 14, the mice were euthanized and the quadriceps muscle in each animal was harvested for bench-work investigation.

### 2.5. Western Blot Analysis

The methodology for Western blot analysis was based on our recent reports [33,34]. In detail, equal amounts (50 μg) of protein extracts were segregated by SDS-PAGE. After electrophoresis, the segregated proteins were transferred onto a polyvinylidene difluoride (PVDF) membrane. Nonspecific sites were blocked by incubation of the membrane in blocking buffer [5% nonfat dry milk in T-TBS (TBS containing 0.05% Tween 20)] at room temperature for one hour. Then the membranes were incubated with the indicated primary antibodies [vascular endothelial growth factor receptor 2 (VEGFR2) (1:1000, Abcam, Kaohsiung, Taiwan), epidermic growth factor receptor (EGFR) (1:1000, Cell Signaling, Kaohsiung, Taiwan), von Willebrand factor (vWF) (1:1000, Abcam), vascular endothelial growth factor (VEGF) (1:1000, Abcam), Ras (1:1000, Abcam), c-Raf (1:1000, Cell Signaling), phosphorylated (p)-MEK1/2(1:1000, Cell Signaling), p-ERK1/2 (1:1000, Sigma-Aldrich, Kaohsiung, Taiwan), total PI3K (1:5000, Abcam), p-PI3K (1:1000, Cell Signaling), total Akt (1:1000, Cell Signaling), p-Akt (1:1000, Cell Signaling), total m-TOR (1:1000, Cell Signaling), p-m-TOR (1:1000, Cell Signaling), p-FAK (1:1000, Cell Signaling), p-Src (1:1000, Cell Signaling), stromal cell-derived growth factor (SDF)-1α, CXCR4 (1:1000, Abcam), and β-actin (1:10000, Chemicon, Billerica, MA, USA)] for one hour at room temperature. Horseradish peroxidase-conjugated anti-rabbit immunoglobulin IgG (1:2000, Cell Signaling) was used as a secondary antibody for one-hour incubation at room temperature.

### 2.6. Immunohistochemical (IHC) and Immunofluorescent (IF) Stains

The protocol of IF staining has been described in our previous reports [34,35]. For IHC and IF staining, rehydrated paraffin sections were first treated with 3% H_2_O_2_ for 10 min and incubated with Immuno-Block reagent (BioSB, Santa Barbara, CA, USA) for 30 min at room temperature. Sections were then incubated with primary antibodies specifically against alpha smooth muscle actin (α-SMA) (1/400, Sigma-Aldrich), CD31 (1/100, Bio-Rad, Kaohsiung, Taiwan), von Willebrand factor (vWF) (1/200, Sigma-Aldrich) and vascular endothelial cell (VEGF) (1:500, Abcam). Sections incubated with the use of irrelevant antibodies served as controls. Three sections of quadriceps specimen from each mouse were analyzed. For quantification, three randomly selected HPFs (200× or 400× for IHC and IF studies) were analyzed in each section. The mean number of positive-stained cells per HPF for each animal was then determined by summation of all numbers divided by nine.

### 2.7. Statistical Analysis

Quantitative data were presented as mean ± standard deviation (SD). Statistical analysis was carried by analysis of variance (ANOVA) followed by Bonferroni multiple comparison post hoc test. SAS statistical software for Windows version 8.2 was utilized. A *p* value < 0.05 was considered statistically significant.

## 3. Results

### 3.1. The Angiogenesis and Protein Expressions of EGFR/VEGFR2, and Cell Proliferation/Growth/Survival and Cell Motility Signaling Pathways 

The in vitro study comprised several groups: the HUVECs were categorized into group A [HUVECs (i.e., as control group)], group B [HUVECs + ECSW (energy stepwise increased from 0.1, 0.14 to 0.2 mJ/mm^2^/140 impulses)], and group C [HUVECs + ECSW/0.20 mJ/mm^2^/140 impulses) + Avastin (20 μM)].

To elucidate the influence of ECSW therapy on angiogenesis, the Matrigel assay was utilized. The results showed that the angiogenesis parameters, i.e., tubular, cluster and network formation, were significantly and progressively higher in group B than in group A (i.e., control group) as the ECSW energy was stepwise increased (Figure 1). However, these increments in the angiogenesis markers were found to be significantly suppressed in group C, suggesting that Avastin inhibited angiogenesis (Figure 1).

Next, to assess whether the ECSW therapy would upregulate the expressions of EGFR and VEGFR2, Western blot analysis was performed. As we expected, when compared to group A, the protein expressions of EGFR and VEGFR2 were significantly increased in group B (Figure 2). However, the protein expressions of these two parameters were significantly downregulated by Avastin treatment (Figure 2). Additionally, the protein expressions of vWF, an indicator of endothelial cell surface marker, and VEGF, an index of angiogenesis, were significantly increased in group B compared to group A, and were significantly reversed in group C (Figure 2).

Furthermore, to clarify what cellular signalings were elicited after ECSW therapy, the Western blot was analyzed once again. The result demonstrated that the protein expressions of cell proliferation (Ras/c-Raf, p-MEK1/2, p-ERK1/2), cell proliferation/growth/survival (p-PI3K/p-Akt/p-m-TOR) and cell motility/proliferation (p-FAK/p-Scr) signaling pathways were significantly increased in group B as compared with group A, and were significantly reversed in group C (Figure 3), suggesting that ECSW therapy enhancement of angiogenesis might be via upregulating these signaling pathways.

### 3.2. Ischemic to Normal Blood Flow (INBF) Ratio Measured by Laser Doppler Scan at Days 1, 7 and 14 after Left CLI Induction

By day 1 after CLI induction, laser Doppler examination demonstrated a significantly higher INBF ratio in group 1 (SC) than in groups 2 (CLI), 3 (CLI + ECSW) and 4 (CLI + ECSW + Avastin), but there was not any obvious difference among the latter three groups (Figure 4). By days 7 and 14 after induction of CLI, the ratio of INBF was highest in group 1, lowest in group 2, and significantly higher in group 3 than in group 4 (Figure 4).

### 3.3. The Protein Expressions of Cell Growth/Angiogenesis Receptors and Angiogenic Factors in CLI Quadriceps Muscle by Day 14 after CLI Induction

First, to assess whether ECSW therapy would activate the expressions of angiogenesis/cell growth receptors and angiogenesis biomarkers, we utilized the tool of Western blot. The result demonstrated that the protein expressions of EGFR and VEGFR2, two indices of cell growth/angiogenesis receptors, were significantly higher in group 3 than in groups 1, 2 and 4, and significantly higher in groups 2 and 4 than in group 1, but they showed no difference between groups 2 and 4 (Figure 5). Additionally, the protein expressions of VEGF, SDF-1α and CXCR4, three indicators of angiogenesis biomarkers, displayed an identical pattern of VEGFR2, whereas the protein expression of vWF, an indicator of endothelial cell surface marker, exhibited an opposite pattern of VEGFR2 among the groups (Figure 5).

### 3.4. The Protein Expressions of Cell Proliferation/Growth/Survival and Cell Motility Signalings in CLI Quadriceps Muscle by Day 14 after CLI Induction

Next, based on our in vitro results, we planned to investigate whether ECSW therapy would also activate the cell proliferation and cell motility signalings in the in vivo study. As expected, the protein expressions of Ras, c-Raf, p-MEK1/2, and ERK1/2, four indicators of cell proliferation/survival signaling parameters, were significantly higher in group 3 than in other groups, significantly lower in group 1 than in groups 2 and 4, but similar between the latter two groups (Figure 6). Consistently, the protein expressions of p-FAK and p-Scr, two unique biomarkers of cell motility, displayed an identical pattern of Ras among the groups (Figure 7). In additional, the ratio of protein expressions of phosphorylated (p)-PI3K/p-Akt/p-m-TOR to the total PI3K/Akt/m-TOR, other biomarkers of proliferation/growth/survival signaling, also displayed an identical pattern of Ras (Figure 7). Taken together, our findings implicated that Avastin could inhibit therapeutic function of ECSW.

### 3.5. Cellular Expressions of Angiogenesis in CLI Quadriceps Muscle by Day 14 after CLI Induction

To evaluate whether the expression in cell level would be comparable with the protein level of angiogenesis, we utilized the IF microscopic instrument. As expected, the cellular expressions of CXCR4 and VEGF, two indicators of endothelial progenitor cell (EPC)/angiogenesis biomarkers, were significantly higher in group 3 than in groups 1, 2 and 4, and significantly higher in groups 2 and 4 than in group 1, but they showed no difference between groups 2 and 4 (Figure 8).

### 3.6. The Microscopic Findings for Identification of Small Vessel Density and the Endothelial Cell Surface Marker in CLI Quadriceps Muscle by Day 14 after CLI Induction

The number of small vessels (i.e., defined as diameter ≤25.0 μM) was significantly higher in group 1 than in other groups and significantly higher in group 3 than in groups 2 and 4, but it did not differ between the latter two groups (Figure 9). Additionally, the IF microscopic finding demonstrated that the number of CD31 cells, an indicator of endothelial cells, was highest in group 1, lowest in group 2 and significantly higher in group 3 than in group 4 (Figure 9).

## 4. Discussion

This study which investigated the therapeutic impact of ECSW on salvaging CLI yielded several striking preclinical implications. First, as compared with the CLI only group, the ratio of INBF (i.e., ratio of blood flow in CLI to normal limb) was remarkably improved in those CLI animals treated by ECSW, indicating that this therapy effectively salvaged the CLI in mice. Second, in vivo studies demonstrated that the salvage of CLI with ECSW therapy was mainly via neovascularization and angiogenesis effects. Third, in vitro and in vivo findings supported the idea that the underlying mechanism of ECSW therapy on enhancing angiogenesis and salvaging the CLI was related to three signaling pathways, including cell proliferation/growth/survival and cell motility.

Interestingly, our studies have previously demonstrated that ECSW therapy significantly preserved the ischemic related organ dysfunction in small and large animals [22,23,24,25,26,27,32]. These studies have further demonstrated that the salvage of ischemic related organ dysfunction in these small and large animals was mainly through enhancing the angiogenesis and SDF-1 angiogenic factors which attracted the EPC mobilization from circulation into the ischemia zone (i.e., a homing phenomenon) for angiogenesis [22,23,24,25,26,27,32], resulting in an increase of the blood flow in the ischemic region. The most important finding in the present study was that as compared with CLI animals without treatment, the blood flow in the CLI area and the molecular and cellular levels of angiogenic biomarkers were substantially increased in the CLI animals after receiving the ECSW treatment. Our findings corroborated the findings of our previous studies [22,23,24,25,26,27,32]. 

When looking at the results of our previous studies [22,23,24,26,27,32,35], we found that the limitations of these studies were that the exact underlying mechanisms for how the ECSW could salvage the CLI [23,32,35] and improve ischemic related heart function [22,24,26,27] have not been fully investigated. An essential finding in the in vitro study was that the ECSW promoted EGFR and VEGFR2 activation in the HUVECs. In additional, when we looked at the result of our in vivo study, we also found that the protein expressions of these two parameters were upregulated in CLI after ECSW treatment, suggesting the results in the in vitro and in vivo studies were comparable. Intriguingly, our previous study [32] has identified ECSW enhanced angiogenesis through VEGFR2 activation and recycling. Accordingly, our findings were comparable with the result from our previous study [32].

The essential findings of the present study, both in the in vitro and in vivo studies, demonstrated that the protein expressions of Ras/c-Raf/MEK/ERK (i.e., cell proliferation/survival pathway), PI3K/Akt/m-TOR (i.e., the signaling for cell proliferation/growth/survival) and p-FAK/p-Scr (i.e., cell motility/proliferation pathway) were significantly increased in the CLI group and further significantly in the CLI + ECSW group as compared to the control group, suggesting that upregulation of these signaling pathways through intrinsic response to stress/ischemic stimulation was remarkably augmented by ECSW therapy. On the other hand, the activation of these signalings was ameliorated by Avastin treatment, implicating that these downstream signalings activated by VEGFR and EGFR could be specifically blocked by the antiangiogenic monoclonal antibody, resulting in lowering the blood flow in the CLI area.

Some findings of this present study should be clarified to avoid misreading by the readers. First, in view of the Matrigel assay and Western blotting, we found that the protein and cellular levels of angiogenesis were inconsistent in terms of Avastin treatment. We suggested that this could be due to the Matrigel assay being focused on only morphological features through semi-quantitative analysis. Thus, it may not be as accurate as the Western blot analysis. Second, when we looked at the in vivo results, we found that angiogenesis biomarkers and protein levels of the aforementioned three cell signaling pathways were notably increased in the CLI group as compared to the SC group. We proposed that it could be an intrinsic response of cells/tissue in the CLI area to ischemic stimulation, especially in a situation of partial loss of microvasculature (i.e., a limited blood flow supplied to the ischemic cells/tissues).

### Study Limitations

This study has limitations. First, the ECSW energy applied to mice was only based on our previous reports [23,28] without assessing the effect of stepwise increase in the dosage of ECSW on the CLI area. Therefore, the optimal in vivo energy of ECSW for improving the blood flow and salvaging the CLI in mice is currently unclear. Second, although the results were attractive and promising, the study period was only 14 days. Therefore, the long-term effect of ECSW therapy on CLI remains uncertain. Third, although extensive work has been performed, we cannot completely rule out the possibility that other signaling pathways may be involved in the angiogenesis. Accordingly, given the findings of the present study, we schematically illustrated the underlying mechanism of ECSW therapy on eliciting the signaling pathways that participate in angiogenesis and salvage of CLI in Figure 10. Finally, this study did not test the stepwise increase in the concentration of Avastin on the impact of cell viability and molecular-cellular levels of angiogenesis. Thus, we did not recommend what was the optimal dosage of Avastin for the in vitro and in vivo studies. Appendix A demonstrated how we assessed angiogenesis suppressed by Avastin in an ex vivo study of rat aortic ring. Appendix A showed cell viability progressively reduced with time after exposure to different dose of Avastin in MTT assay. 

In conclusion, the results of the present study showed that ECSW therapy promoted blood flow in CLI and salvaged the ischemic limb leg in mice mainly through activating the cell proliferation, growth and motility signalings, followed by enhancing angiogenesis.

## Figures and Tables

**Figure 1 biomedicines-10-00117-f001:**
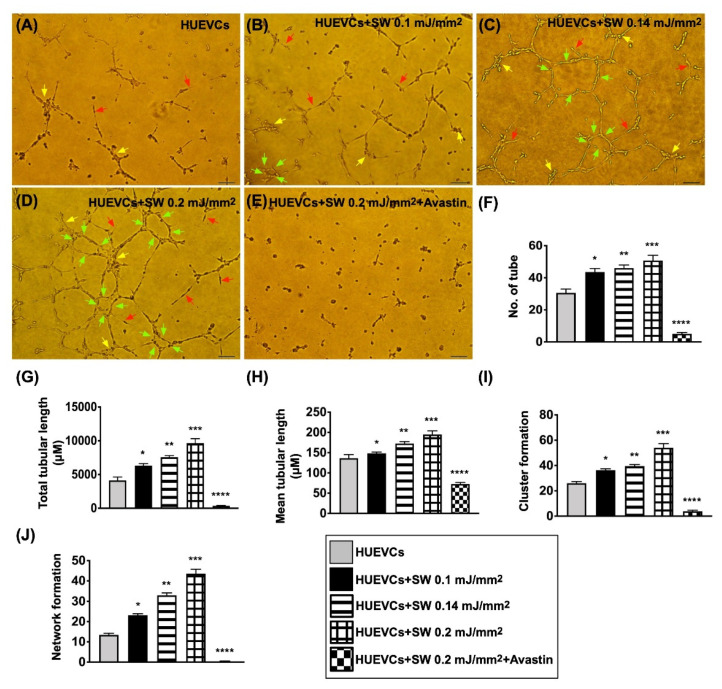
Impact of stepwise increased ECSW energy and Avastin on angiogenesis. (**A**–**E**) Illustrating the morphological features of Matrigel assay for identification of stepwise increase in ECSW energy on enhancing angiogenesis in human umbilical vein endothelial cells (HUVECs). The parameters of angiogenesis, including: (1) tubular formation (red arrows), (2) cluster formation (yellow arrows) and (3) network formation (green color). (**F**–**J**) Statistical analysis for angiogenesis parameters (F: number of tubules; G: total tubular length; H: mean tubular length; I: cluster formation; J: network formation). “*” represents in comparison with the control, * for *p* < 0.05, ** for *p* < 0.01, *** for *p* < 0.001, **** for *p* < 0.0001. Scale bar in right lower corner represents 50 µM. *n* = 6 for each group. HUVECs = human umbilical vein endothelial cells; SW = shock wave.

**Figure 2 biomedicines-10-00117-f002:**
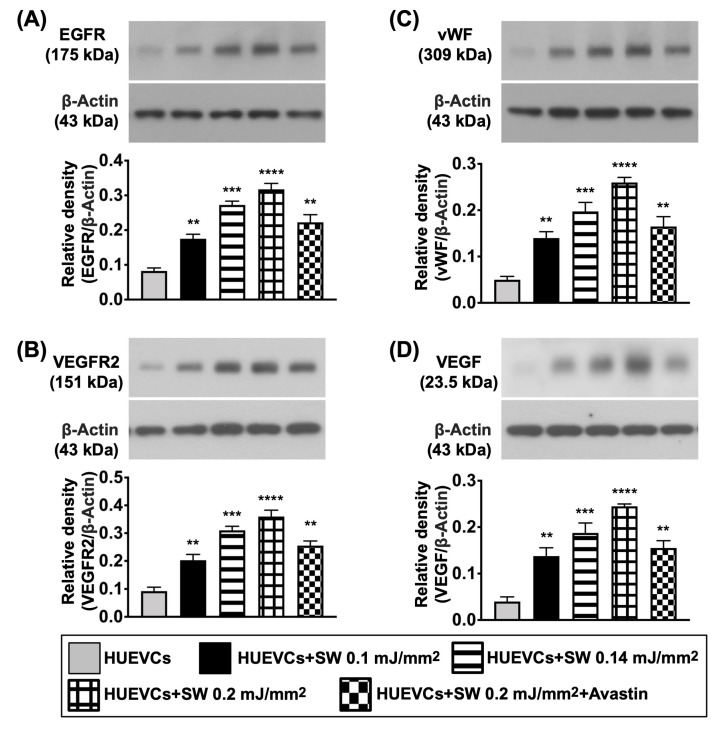
ECSW therapy upregulated the protein expressions of EGFR and VEGFR2, and angiogenic biomarkers in HUVECs. (**A**–**D**) Western blot analyses showed the results of protein expressions of epidermal growth factor receptor (EGFR) (**A**), vascular endothelial growth factor receptor 2 (VEGFR2) (**B**), von Willebrand factor (vWF) (**C**) and vascular endothelial growth factor (VEGF) (**D**). “*” represents in comparison with the control, ** for *p* < 0.01, *** for *p* < 0.001, **** for *p* < 0.0001. *n* = 4 for each group. HUVECs = human umbilical vein endothelial cells; SW = shock wave.

**Figure 3 biomedicines-10-00117-f003:**
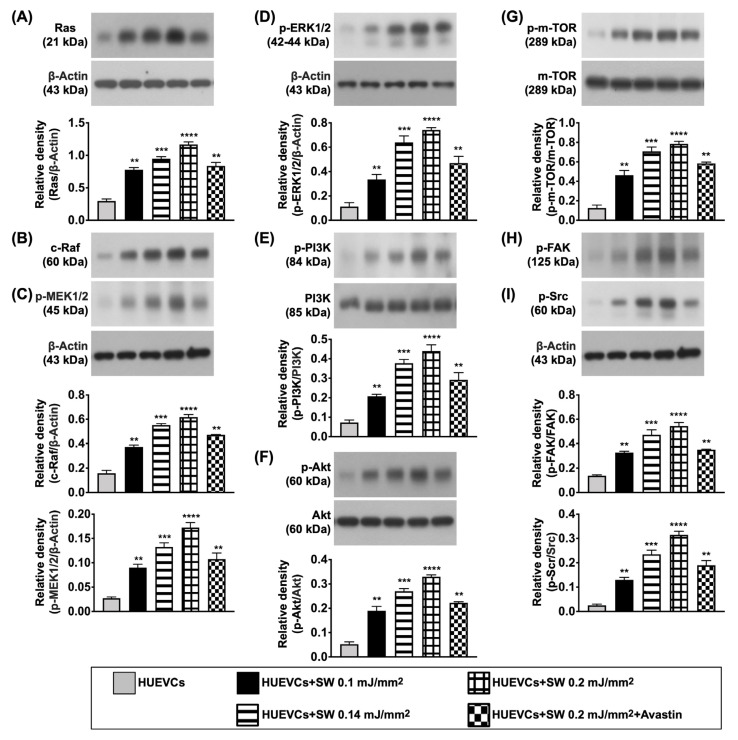
Protein expressions of cell proliferation/growth/survival, and cell motility signalings. (**A**–**I**) Western blot analyses showed the results of protein expressions of Ras (**A**), c-Raf (**B**), phosphorylated (p)-MEK1/2 (**C**), p-ERK1/2 (**D**), p-PI3K (**E**), p-Akt (**F**), p-m-TOR (**G**), p-focal adhesion kinase (FAK) (**H**) and p-Scr (**I**). “*” represents in comparison with the control, ** for *p* < 0.01, *** for *p* < 0.001, **** for *p* < 0.0001. *n* = 4 for each group. HUVECs = human umbilical vein endothelial cells; SW = shock wave.

**Figure 4 biomedicines-10-00117-f004:**
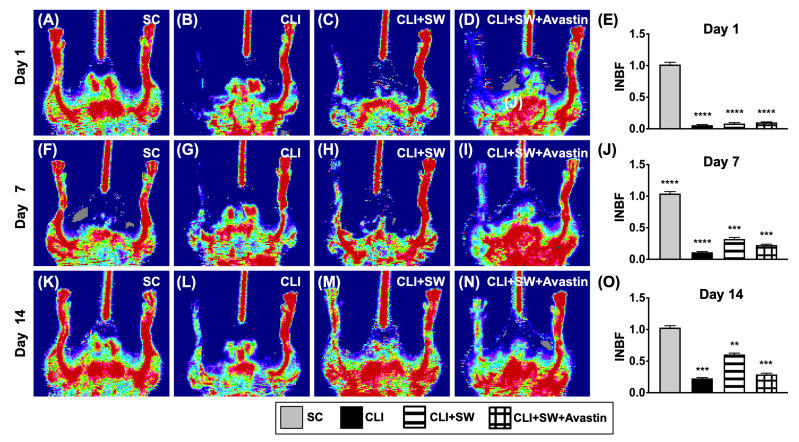
Ischemic/normal blood flow (INBF) ratio measured by laser Doppler scan at days 1, 7 and 14 after left CLI induction. (**A**–**D**) Illustrating the laser Doppler finding of ratio of left limb (ischemia) to right limb (normal) blood flow (i.e., INBF) at day 1 after CLI procedure among the four groups. (**E**) Analytical result of ratio of INBF, **** for *p* < 0.0001. (**F**–**I**) Illustrating the laser Doppler finding of ratio of INBF at day 7 after CLI procedure among the four groups. (**J**) Analytical result of ratio of INBF, *** for *p* < 0.001, **** for *p* < 0.0001. (**K**–**N**) Illustrating the laser Doppler finding of ratio of INBF at day 14 after CLI procedure among the four groups. (**O**) Analytical result of ratio of INBF, ** for *p* < 0.01, *** for *p* < 0.001, *n* = 6 for each group. SC = sham-operated control; SW = shock wave.

**Figure 5 biomedicines-10-00117-f005:**
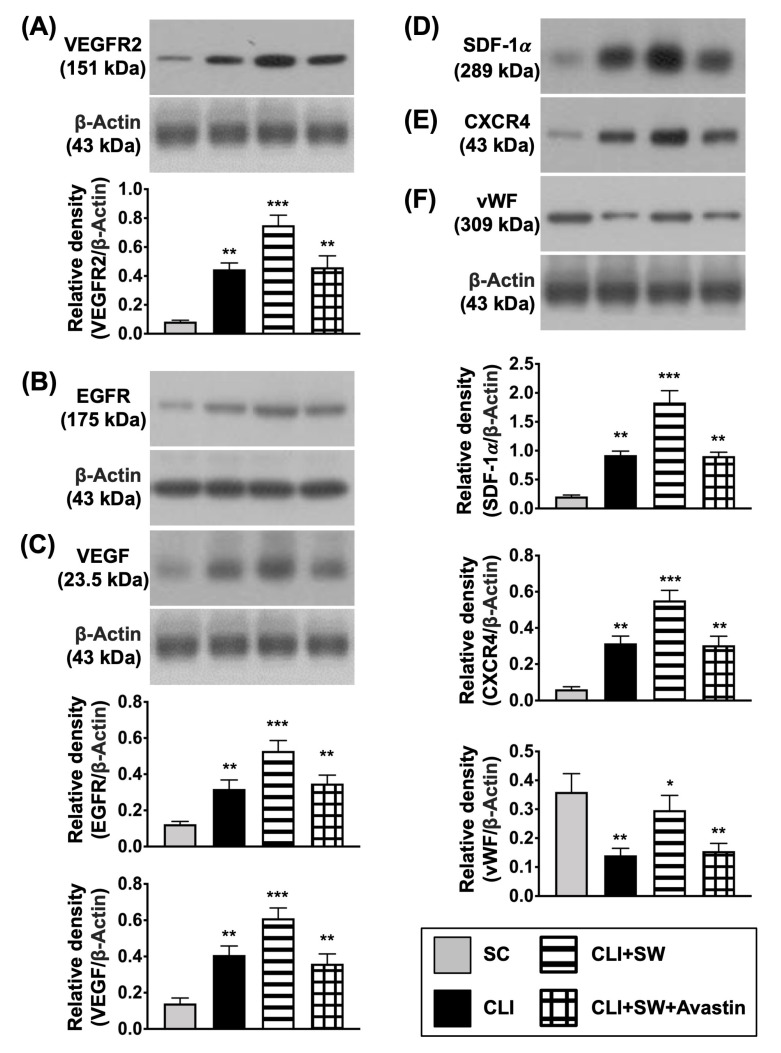
The protein expressions of cell growth, angiogenesis receptor and angiogenic factors in CLI quadriceps muscle by day 14 after CLI induction. (**A**–**F**) Western blot analyses showed the results of protein expressions of vascular endothelial growth factor receptor 2 (VEGFR2) (**A**), epidermal growth factor receptor (EGFR) (**B**), vascular endothelial growth factor (VEGF) (**C**), (**D**) stromal cell derived factor (SDF)-1α, CXCR4 (**E**), and von Willebrand factor (vWF) (**F**), * for *p* < 0.05, ** for *p* < 0.01, *** for *p* < 0.001. *n* = 6 for each group. SC = sham-operated control; SW = shock wave.

**Figure 6 biomedicines-10-00117-f006:**
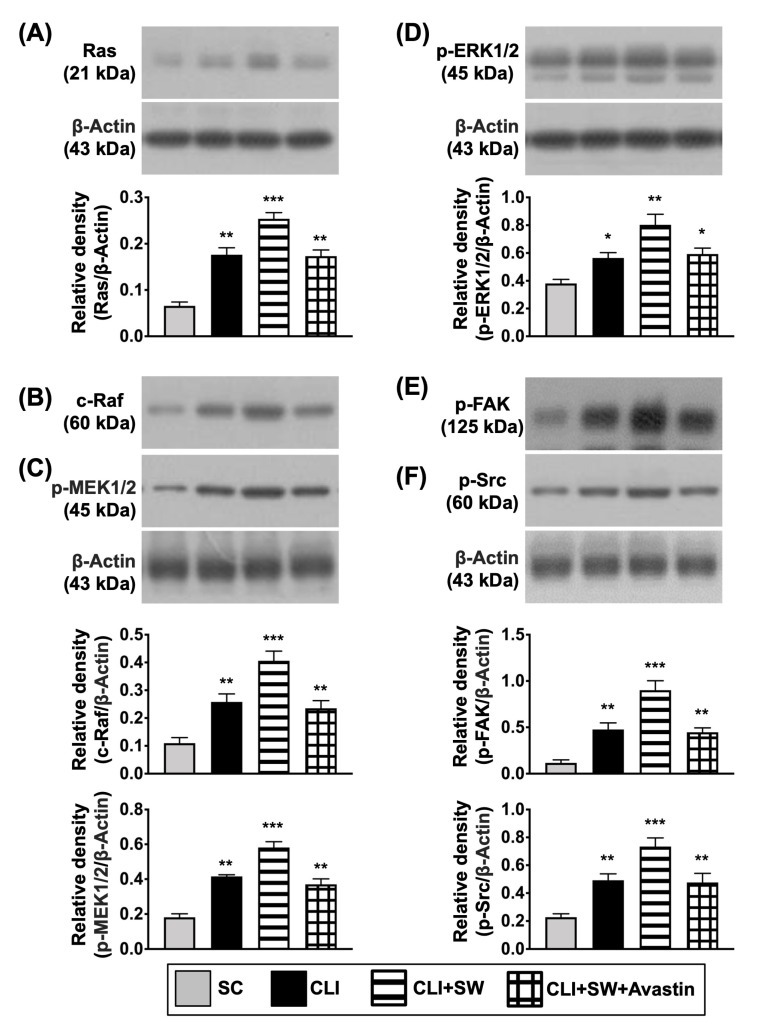
The protein expressions of cell proliferation and cell mobility signalings in CLI quadriceps muscle by day 14 after CLI induction. (**A**–**F**) Western blot analyses showed the results of protein expressions of Ras (**A**), c-Raf (**B**), phosphorylated (p)-MEK1/2 (**C**), p-ERK1/2 (**D**), p-focal adhesion kinase (p-FAK) (**E**) and p-Scr (**F**), * for *p* < 0.05, ** for *p* < 0.01, *** for *p* < 0.001. *n* = 6 for each group. SC = sham-operated control; SW = shock wave.

**Figure 7 biomedicines-10-00117-f007:**
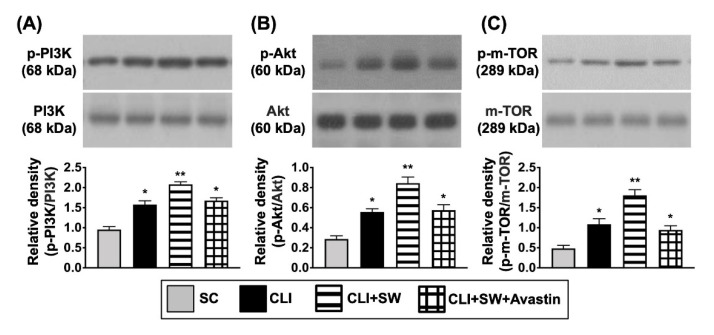
The protein expressions of cell proliferation/growth/survival signaling in CLI quadriceps muscle by day 14 after CLI induction. (**A**–**C**) Western blot analyses showed the results of protein expressions of phosphorylated (p)-PI3K (**A**), p-Akt (**B**) and p-m-TOR (**C**). * for *p* < 0.05, ** for *p* < 0.01. *n* = 6 for each group. SC = sham-operated control; SW = shock wave.

**Figure 8 biomedicines-10-00117-f008:**
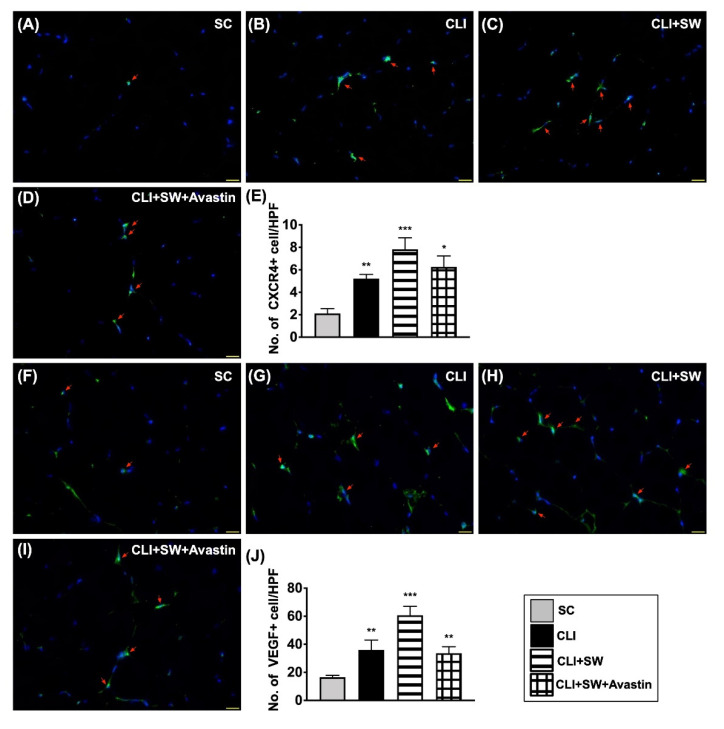
Cellular expressions of angiogenesis in CLI quadriceps muscle by day 14 after CLI induction. (**A**–**D**) Illustrating the immunofluorescent (IF) microscopic finding (400×) for identification of cellular expression of CXCR4 (green color, red arrows). (**E**) Analytical result of number of CXCR4+ cells, * for *p* < 0.05, ** for *p* < 0.01, *** for *p* < 0.001. (**F**–**I**) Illustrating the IF microscopic finding (400×) for identification of cellular expression of vascular endothelial growth factor (VEGF) (green color, red arrows). (**J**) Analytical result of number of VEGF+ cells, ** for *p* < 0.01, *** for *p* < 0.001. All scale bars in right lower corner represent 20 µM. HPF = high-power field. *n* = 6 for each group. SC = sham-operated control; SW = shock wave.

**Figure 9 biomedicines-10-00117-f009:**
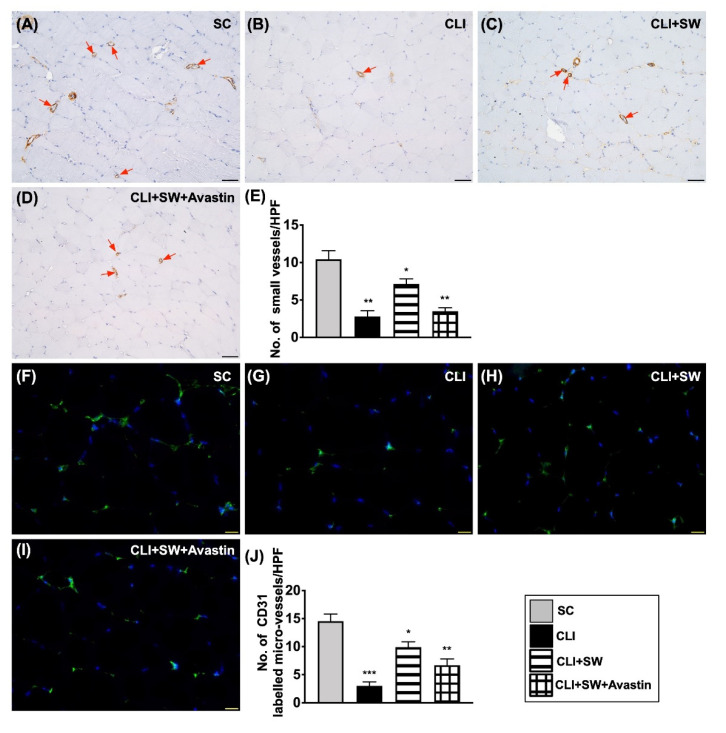
Small vessel density and the endothelial cell surface marker in CLI quadriceps muscle by day 14 after CLI induction. (**A**–**D**) Illustrating microscopic finding (200×) of alpha positively-stained smooth muscle actin (α-SMA) for identifying number of small vessels (i.e., defined as diameter ≤25.0 μM) (gray color) (red arrows). (**E**) Analytical result of number of small vessels, * for *p* < 0.05, ** for *p* < 0.01. All scale bars in right lower corner represent 50 µM. (**F**–**I**) Illustrating the immunofluorescent microscopic finding (400×) for identification of CD31+ cells (green color). (**J**) Analytical result of number of CD31+ cells, * for *p* < 0.05, ** for *p* < 0.01, *** for *p* < 0.001. All scale bars in right lower corner represent 20 µM. HPF = high-power field. *n* = 6 for each group. SC = sham-operated control; SW = shock wave.

**Figure 10 biomedicines-10-00117-f010:**
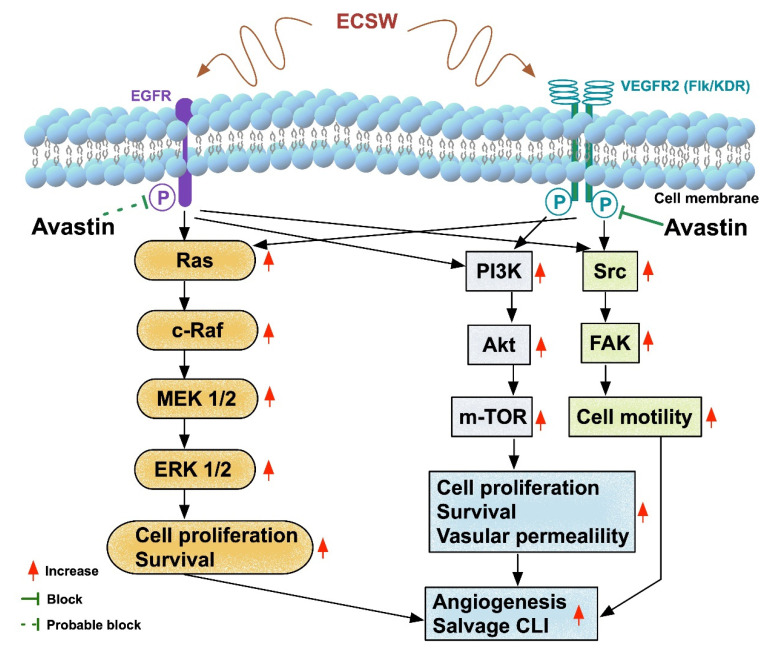
Schematical illustration of the underlying mechanism of ECSW therapy on salvaging CLI in mice. ECSW = extracorporeal shock wave; EGFR = epidermic growth factor receptor; VEGFR = vascular endothelial growth factor receptor; CLI = critical limb ischemia.

## Data Availability

The datasets of the present study are available from the corresponding author upon request.

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
