# Peer review of "Extracorporeal Shock Wave Therapy Salvages Critical Limb Ischemia in B6 Mice through Upregulating Cell Proliferation Signaling and Angiogenesis"

_biomedicines, 2022, doi:10.3390/biomedicines10010117_

Round 1

Reviewer 1 Report

The authors presented the usefulness of shock wave therapy for ischemic. Each data showed significant differences and reliability. Generally, the manuscript is very clear and interesting. However, there are some small points that need to improve. Please see below and deal with them.

  1. In Figure1(G),(H), The author needs to show the unit of the Y axis.
  2. There are so many unnecessary ”-" in the text. For example, " pa-rameters" in line 195, "ar-rows" in line 196, "mul-tiple" in line 219 etc.. Please do spell check and repair the text. ã€€ 

Author Response

Response to Reviewer's Comments (Reviewer #1)

Dear Reviewer:

Your constructive criticism is greatly appreciated. We have made the following responses to comply with your honorable suggestions:

Response to Comments and Suggestions for Authors

Comment 1: In Figure1(G),(H), The author needs to show the unit of the Y axis.

Response 1: Yes, according to your recommendation, we have provided the unit of the Y-axis (μM: micrometer) in our revised Figure 1.

Comment 2: There are so many unnecessary ”-" in the text. For example, " pa-rameters" in line 195, "ar-rows" in line 196, "mul-tiple" in line 219 etc.. Please do spell check and repair the text. 

Response 2: Yes, this unnecessary “-” in the text has been deleted in our revised manuscript.

We would like to take this opportunity to express our appreciation for your detailed review of the article and the kindness of giving us valuable suggestions. Thank you very, very much!

Reviewer 2 Report

The manuscript submitted by Sung et al. presents the continuation of extracorporeal shock wave therapy study. Although the study is quite well designed and shows some new aspects, the presentation of this paper is very confusing and contains a lot of editorial, grammar and logic flaws. Before submission to the journal, everything should be double-checked to avoid a bad first impression. The submission/paper is our business card in the scientific area.

The comments are as follows:

  1. Since all of the presented pathways are connected, it is a bit overexpressed to use some of the protein as markers of specific cellular pathway e.g. PI3K, Akt are not only involved in cell stress but even much more in cell proliferation, survival, growth pathways. The division should be removed or presented in a much milder way.
  2. How to explain the dramatic effect of Avastin on HUVEC angiogenesis? The cells treated with Avastin respond much stronger even than control cells but this effect is not reflected in western blotting data and other analyses. How to explain this discrepancy? Why did CLI induction cause pEGFR, pErk, pFAK, pSrc pMEK, pPI3K, VEGF, SDF 1a activation? Since the Avastin treatment restored a similar phosphoprotein expression level like CLI, that is unclear what is the mechanism of Avastin action in the cells. It should be elucidated in the very enigmatic discussion section. Why the authors have used 20 μM Avastin concentration? Was this effect dose-dependent?
  3. The legends of the figures should be modified to present the description consistently and compactly, not repeat the same unclear sentences. What is marked in blue in Fig 8? What does it mean: “Analytical result of number of VEGF+ cells” – line 320, 322?
  4. The statistical analysis is unclear. In many places, the authors claim that some changes are statistically important but it is not marked in the plot. What are the p-values for *, †, ‡, §, ¶)? I think much better is using standards: * for p<0.05, ** for p<0.01, *** for p<0.001. Moreover, the p-value for comparison to other groups (not only to SC or HUVEC) is unmarked.
  5. Figures 6 and 7 should be combined since they present similar results.
  6. I propose to remove the figure numbers in the titles of results sections.
  7. marker protein ladder lines should be marked on the western blotting results, not just the predicted mass
  8. The text contains many illegal hyphenation, double spaces, combined words without spacesg line 264, and typos e.g. lines 397 and 272 (“ frist”)
  • In vitro/ in vivo as a Latin term should be italicized
  • Since western blotting name is not related to the name of discoverer like Northern blotting, it should be written in small letter
  • It is unclear why antiplatelet is written together but anti-ischemic has a dash between words

Author Response

Response to Reviewer's Comments (Reviewer #2)

Dear Reviewer:

Your constructive criticism is greatly appreciated. We have made the following responses to comply with your honorable suggestions (Note: The revised parts of the manuscript in response to Reviewer’s comments have been marked in red color):

Response to reviewer’s comments to author

Comment 1: Since all of the presented pathways are connected, it is a bit overexpressed to use some of the protein as markers of specific cellular pathway e.g. PI3K, Akt are not only involved in cell stress but even much more in cell proliferation, survival, growth pathways. The division should be removed or presented in a much milder way

Response 1: Yes, according to your recommendation, we have rewritten these inappropriate descriptions in our revised manuscript.

Comment 2:

Comment 2-A: How to explain the dramatic effect of Avastin on HUVEC angiogenesis? The cells treated with Avastin respond much stronger even than control cells but this effect is not reflected in western blotting data and other analyses.

Response 2-A: Dear reviewer, we would like to reply to your criticism as the following: (1) In the in vitro study, we utilized the concentration of Avastin was based on the previous reports (5-10 μM from Turk J Pharm Sci 2019 Sep;16(3):303-309;  0.25 mg/Ml from Clin Exp Ophthalmol 2015;43(2):173-9.) with some modification. (2) In view of Matrigel assay, we suggest that the cells treated with Avastin respond much stronger even than that of the control cells could be reasonable due to the control cells without any treatment. (3) As we know that the Matrigel assay only a morphological feature of semi-quantitative analysis. Thus, it may not be accurate as the Western blot analysis. Perhaps, this could at least in part for explaining your question. Anyway, our in vitro study proved our hypothesis that the Avastin really suppressed the angiogenesis of endothelial cells.

Comment 2-B: How to explain this discrepancy? Why did CLI induction cause pEGFR, pErk, pFAK, pSrc pMEK, pPI3K, VEGF, SDF 1a activation? Since the Avastin treatment restored a similar phosphoprotein expression level like CLI, that is unclear what is the mechanism of Avastin action in the cells. It should be elucidated in the very enigmatic discussion section.

Response to 2-B: (1) We suggest that it could be an intrinsic response of cells/tissue in CLI area to ischemic stimulation, especially in a situation of non-total loss of microvasculature (i.e., supplied only by limited blood flow to the ischemic cells/tissues) (2) In our results, we did not find that the Avastin could restore “a similar phosphoprotein expression level like CLI”. Based on the results of our study, we had schematically illustrated the underlying mechanism of ECSW therapy (refer to Figure 10) on salvaging the CLI and Avastin served as a strong inhibitor of angiogenesis, resulting in against the ECSW-induced angiogenesis/neovascularization. (3) Yes, according to your recommendation, we have discussed this issue in the Discussion Section of our revised manuscript.   

Comment 2-C: Why the authors have used 20 μM Avastin concentration? Was this effect dose-dependent?

Response 2-C: As we had just mentioned in Response 2-A “In the in vitro study, we utilized the concentration of Avastin was based on the previous reports (5-10 μM from Turk J Pharm Sci 2019 Sep;16(3):303-309;  0.25 mg/Ml from Clin Exp Ophthalmol 2015;43(2):173-9.). We are honest to tell you that we did not test the impact of stepwise increased concentration of Avastin on cell viability or on the cellular-molecular levels of angiogenesis in the present study. We know that this is the limitation of our study that has been discussed on the Limitation paragraph of revised manuscript. 

Comment 3: The legends of the figures should be modified to present the description consistently and compactly, not repeat the same unclear sentences. What is marked in blue in Fig 8? What does it mean: “Analytical result of number of VEGF+ cells” – line 320, 322?

Response 3: Yes, according to your recommendation, we have rewritten the description of Figure legends in our revised manuscript. RegaRding the Figure 8, we have added the arrows to identify the expression of the positive stain of VEGF in CLI area.

Comment 4: The statistical analysis is unclear. In many places, the authors claim that some changes are statistically important but it is not marked in the plot. What are the p-values for *, †, ‡, §, ¶)? I think much better is using standards: * for p<0.05, ** for p<0.01, *** for p<0.001. Moreover, the p-value for comparison to other groups (not only to SC or HUVEC) is unmarked.

Response 4: Yes, according to your recommendation, we have rewritten the expression of statistical analyses in our revised manuscript.

Comment 5: Figures 6 and 7 should be combined since they present similar results.

Response 5: Dear reviewer, we are honest to tell you that prior to submission of our manuscript to your journal, we had really combined Figures 6 and 7 together as one Figure. However, we found that it became more condensed/crowded and unclearly to be read due to the Figure was too big. Thus, we beg you to consider these two Figures are not combined as one.   

Comment 6: I propose to remove the figure numbers in the titles of results sections.

Response 6: Yes, according to your recommendation, we have deleted the figure numbers in the titles of results sections in our revised manuscript.

Comment 7: marker protein ladder lines should be marked on the western blotting results, not just the predicted mass

Response 7: Dear reviewer, to simplify the expression of Western blot analysis, we only left the “predicted mass”, i.e., the protein band. However, these “marker protein ladder lines” have been reserved in the raw materials of Western blotting.  

Comment 8: The text contains many illegal hyphenation, double spaces, combined words without spacesg line 264, and typos e.g. lines 397 and 272 (“ frist”)

Response 8: Yes, according to your comment, we have another English speaker to edit our manuscript again.

Response to minor comments

Comment 1: In vitroin vivo as a Latin term should be italicized

Response 1: Yes, these have been corrected

Comment 2: Since western blotting name is not related to the name of discoverer like Northern blotting, it should be written in small letter

Response 2: Yes, we have rewritten this inappropriate writing in our revised manuscript. 

Comment 3: It is unclear why antiplatelet is written together but anti-ischemic has a dash between words

Response 3: This inappropriate expression has been corrected in our revised manuscript. 

We are greatly indebted to you for your professional comments.

Round 2

Reviewer 2 Report

Although some text changes are incorporated into the manuscript, there are also some modifications which are commented by the authors as “done” but they are not really improved in the paper e.g in study limitations section is still the improper word „frist”, western blotting is written with a capital letter (even in the authors' responses), there is no consequence in usage of words containing “anti” e.g antiplatelet is written all together but anti-ischemic contains a dash between words. Moreover, according to gold standards, at least one band of the marker protein ladder has to be shown on the western blotting results. Predicted mass is just a suspicion but the marker protein ladder shows the real protein mass. Indeed it is really nice to see the original western blotting data in raw material but as I mentioned before it has to be also marked in the main figures.  

That is nice that the authors are aware of the limitations but it does not change the situation. To properly understand the results not only in pharmacokinetics aspects but also to verify the dose-effect relationship it is critical to know what is the effect of increased/decreased Avastin concentration on viability, proliferation, angiogenesis of cells. To keep the high standard of the paper, the Avastin concentration curve should be performed and included in Fig 1 or supplementary data.

Author Response

Response to reviewer’s comments

Round 3

Reviewer 2 Report

The manuscript still contains some typos and flaws but its value is improved in comparison to previous one. Therefore I hope it would be suitable for publication in Biomedicines.